# Genetic and Biochemical Aspects of Floral Scents in Roses

**DOI:** 10.3390/ijms23148014

**Published:** 2022-07-20

**Authors:** Shaochuan Shi, Zhao Zhang

**Affiliations:** 1Vegetable Research Institute, Shandong Academy of Agricultural Science, Jinan 250100, China; shaochuan0066@126.com; 2Beijing Key Laboratory of Development and Quality Control of Ornamental Crops, Department of Ornamental Horticulture, China Agricultural University, Beijing 100193, China

**Keywords:** *Rosa*, floral scents, terpenoids, phenylpropanoids, benzenoids, molecular breeding

## Abstract

Floral scents possess high ornamental and economic values to rose production in the floricultural industry. In the past two decades, molecular bases of floral scent production have been studied in the rose as well as their genetic inheritance. Some significant achievements have been acquired, such as the comprehensive rose genome and the finding of a novel geraniol synthase in plants. In this review, we summarize the composition of floral scents in modern roses, focusing on the recent advances in the molecular mechanisms of floral scent production and emission, as well as the latest developments in molecular breeding and metabolic engineering of rose scents. It could provide useful information for both studying and improving the floral scent production in the rose.

## 1. Introduction

The floral scent is one of the most important traits in plants, which is essential to the fertilization of angiosperm plants by attracting and guiding pollinators [1]. Some volatile compounds in floral scents play important roles in plant defense against detrimental animals [2,3]. To humans, floral scent is an important flower trait; it brings mental pleasure. In addition, floral scent provides essential flavor to the food and perfume industries [4,5].

Most floral scent chemicals are produced by three general metabolic pathways, i.e., terpenoids, phenylpropanoids/benzenoids, and fatty-acid derivatives [1,6]. In plants, terpenes are the largest group of floral scent compounds, which are synthesized by two divergent pathways. One is the 2-C-Methyl-D-Erythritol-4-Phosphate (MEP) pathway, which is mainly located in plastids [7] and is responsible for the production of mono- and diterpenes [8]. The other is the mevalonate (MVA) pathway, which is mainly located in the cytosol, endoplasmic reticulum, and peroxisomes [9,10], and is responsible for the production of volatile sesquiterpenes. Phenylpropanoids/benzenoids represent the second-largest class of floral scent compounds [11], which are exclusively derived from L-phenylalanine (L-Phe). Most phenylpropanoids are not volatile unless they are further acylated or methylated at the C9 position, while benzenoids are volatile, which are synthesized through a branch of the phenylpropanoid pathway, i.e., the cinnamic acid pathway [12]. Fatty acid derivatives constitute the third largest class of flower volatiles, including low-molecular-weight alcohols, aldehydes, and lipids. Biosynthesis of volatile fatty acid derivatives is initiated by stereo-specific oxygenation of unsaturated C18 fatty acids, linolenic, and linoleic, and subsequently catalyzed by the lipoxygenase (LOX) pathway [13,14].

For centuries, roses have been one of the most important crops in the floriculture industry [15], and are highly popular worldwide as garden ornamental plants and cut flowers. Floral scent, as an important characteristic, not only improves the ornamental value of roses, but provides essential fragrances and flavorings for spices, perfumes, and cosmetics in related industries. Additionally, rose essential oil, which is composed of floral scent compounds, can be used as an analgesic or antispasmodic [16,17]. However, the development of fragrances in roses has been at a disadvantage in the breeding program, which has focused on the longevity and visual attributes of cut flowers for centuries [18]. The cause for the loss of fragrance in these flowers remains unknown, but it does not seem to conflict with the increase in the vase life [19]. Unraveling the molecular bases of floral scents is not only a fascinating topic in rose biology, but is also helpful for improving the yield of roses in the floriculture industry.

Here, we will review the current progress of genetic and molecular mechanisms controlling rose scents, including scent composition, production, and emission, and summarize the development of molecular breeding and engineering techniques for the improvement of rose scents.

## 2. Scent Composition of Modern Roses

The evolution of the floral scent is a complex matter, which is influenced by various factors, including the dynamics between biosynthetic pathways and the balanced selection between pollinators and florivores, which may result in relatively rapid evolution [11].

Roses have been cultivated as early as 3000 BC in China, western Asia, and northern Africa [20]. Since the 14th century, when Chinese roses were first introduced to Europe, the Chinese and European roses began to hybridize extensively, forming the genetic basis of the ‘modern rose cultivars’ *Rosa hybrida* [21]. Although *Rosa* comprises approximately 200 species, only 8~20 contribute to the genetic make-up of modern roses, including Chinese rose *R. chinensis, R. multiflora,* and *R. gigantea*, and European rose *R. moschata*, *R. gallica, R. canina,* and *R. Phoenicia* [22,23]. Chinese and European roses differ greatly in scent composition [24,25,26,27,28,29,30]. Chinese roses principally produce lipid-derived alcohols and esters (such as hexenol and hexenyl acetate) and aromatic compounds (such as 3,5-dimethoxytoluene (DMT) and 1,3,5-trimethoxybenzene (TMB)), whereas the major scent components of European roses are 2-phenylethanol (2-PE) and a number of monoterpenes (such as rose oxide, geraniol, and nerol) (Table 1).

Many modern rose flowers have earthy and spicy notes due to the presence of DMT [24,36]. DMT is a product of the phenolic methyl ethers (PME) synthesis pathway and is peculiar to Chinese ancient roses. The DMT synthesis pathway in modern roses is speculated to be obtained from Chinese roses, such as *R. gigantean* [36]. DMT is the basis of the ‘tea scent’ of modern roses. The tea scent is characterized by a combination of phenolic molecules from both Chinese and European lineages, which is reminiscent of ‘black tea’. It is brought to modern roses through an intermediate group of tea and hybrid tea roses, which are derived from the crosses of *R. chinensis* and *R. gigantea* with *R. moschata.* An array of methoxylated phenolics can be produced by these roses, such as DMT, TMB, methyleugenol, and methylisoeugenol, and a variety of alcohols and esters, such as 2-PE, citronellol, geraniol, 2-phenylethylacetate, and geranyl acetate, as well as mono- and sesquiterpenes (predominantly germacrene D), among which, DMT generally represents up to 90% of the total flower volatiles [25,27,28]. *cis*-3-hexenyl acetate and *cis*-3-hexenol are also derived from *R. chinensis*, which give a leafy green note to modern roses [27].

In modern roses, the main components of rose oil, including linalool, citronellol, nerol, and geraniol, are inherited from ancient European roses [27]. The damask rose (*R. damascene*) is an important intermediate, which is a progeny of ancient European roses by the crosses between *R. gallica* and *R. Phoenicia*. Iran is the center of diversity of the damask rose, from where the original oil-bearing cultivars are transferred to Turkey and Bulgaria [37,38,39,40]. The damask rose produces a floral scent characterized by rose oxide, which is rich in alcohols, such as 2-PE, geraniol, and nerol [28,30,41]. Among the components, 2-PE is a dominant aroma compound with a rose-like odor and characterized by the typical floral scent of modern roses [40].

## 3. Molecular Research Progress on Rose Scent Biosynthesis

Molecular and genetic approaches, including the candidate gene approach, transcriptomic analysis, and genetic mapping, have been used to identify scent-related genes and to unravel the gene expression network and floral scent inheritance in roses [31,33,42,43,44,45,46]. With these tools and approaches, dozens of scent-related genes have been identified and functionally validated, and many pathways of floral scent production have been uncovered in the rose (Figure 1 and Table 2).

### 3.1. Biosynthesis of Terpenoids in Rose Floral Scents

Dimethylallyl pyrophosphate (DMAPP) and isopentenyl pyrophosphate (IPP) are synthesized from glyceraldehyde-3-phosphate (G3P) and pyruvate by a series of enzymes in plasmids [11]. As an enzyme for the synthesis of DMAPP and IPP, 1-deoxy-D-xylulose-5-phosphate reductoisomerase (RrDXR) is shown to play a key role in the production of volatile monoterpenes in *R. rugosa* [50]. Geranyl pyrophosphate (GPP) is synthesized from DMAPP and IPP by the MEP pathway in plastids [60]. GPP is a precursor of monoterpenes in plants, from which various monoterpenes are produced by a series of monoterpene synthases. In roses, a novel monoterpene synthase, designated as RhNUDX1, is responsible for geraniol synthesis. Recombinant RhNUDX1 only shows the diphosphohydrolase activity by transforming GPP to geranyl monophosphate (GP) in vitro. However, in a cytosolic context, RhNUDX1 is equivalent to geraniol synthase (GES), which is found in other plants responsible for geraniol production [18]. In rose cultivar *Rosa*× *wichurana*, another NUDX1 gene *RwNUDX1-2* is involved in the biosynthesis of a group of sesquiterpenoids, especially E,E-farnesol. It is proposed that roses utilize different NUDX1 protein complexes to generate different derivatives [61].

Monoterpene alcohols are normally transformed to acetate esters by alcohol acetyltransferase (AAT), by which the acetyl moiety is transferred from acetyl-CoA to the alcoholic substrate [62,63]. In the rose, an AAT, RhAAT 1, is isolated, which shows limited substrate specificity in vitro. Its preferred substrate is geraniol, while it can also accept other alcohols as substrates, including citronellol, nerol, 1-octanol, 2-PE, and *cis*-3-hexen-1-ol [32]. Despite the fact that the preferred substrate is geraniol in vitro, the transgenic petunia flower of *RhAAT1* mainly produces phenylethyl acetate and phenylmethyl acetate using 2-PE and benzyl alcohol as the substrates. When fed with geraniol or octanol, the transgenic flowers also produce acetates, suggesting its dependence on substrate availability in planta [51].

As a sesquiterpene, germacrene is a common ingredient of rose floral scents and an intermediate in the biosynthesis of other sesquiterpenes [64]. In *R. hybrida*, a gene (Clone *FC0592*) for sesquiterpenes has been identified and confirmed to produce germacrene D in an in vitro assay with farnesyl pyrophosphate (FPP) as the substrate [31].

Carotenoids are one class of tetraterpenoids, from which C_13_-norisoprenoids (such as monoterpenes β-damascenone, α-ionone, and β-ionone) are generated by degradation. A carotenoid cleavage (di-)oxygenase (CCD) gene *RdCCD1* was confirmed to be responsible for the accumulation of C_13_-norisoprenoids in flowers of *R. damascene* and the cleavage of a variety of carotenoids in in vitro assays [48]. However, as the subclass 4 of *CCD* genes, *RdCCD4* cannot utilize β-carotene as a substrate in in vivo assays, indicating that it is not involved in the production of β-ionone in *R. damascene* [49].

### 3.2. Biosynthesis of Phenylpropanoids/Benzenoids in Rose Floral Scents

In one branch pathway of phenylpropanoid synthesis, L-Phe is the direct precursor of 2-PE and its β-D-glucopyranoside (2-PEG) [41]. In roses, L-Phe can be converted into phenyl acetaldehyde (PAA) by both aromatic amino acid decarboxylase (AADC) and phenyl acetaldehyde synthase (PAAS) [52,65], and PAA is subsequently converted to 2-PE by phenyl acetaldehyde reductase (PAR) [53,58]. Another 2-PE biosynthetic pathway via phenylpyruvic acid (PPA) from L-Phe has been reported in roses. The aromatic amino acid aminotransferase (AAAT) catalyzes the production of PPA from L-Phe, and RNAi suppression of the AAAT gene *RyAAAT3* decreases 2-PE production in rose protoplasts [54]. In this pathway, phenylpyruvate decarboxylase (RyPPDC) is also shown to be responsible for 2-PE production. However, *RyPPDC,* as a heat adaptation in the summer, is likely to participate in an alternative principal pathway for rose floral scent production [55].

Multiple branching pathways are involved in the synthesis of rose benzenoids. In the branching pathway for TMB, phloroglucinol O-methyltransferase (POMT) catalyzes the first methylation step of phloroglucinol (PLG) to 3,5-dihdroxyanisole (DHA) in *R. chinensis* [56]. DHA is subsequently converted to TMB by two orcinol O-methyltransferases, OOMT1 and OOMT2, through two final methylation reactions [28,35]. OOMT1 and OOMT2 are also responsible for DMT synthesis with orcinol as the initial substrate in the first and the second methylation steps, respectively [28,35]. Despite sharing a 96.5% similarity at the amino acid level, OOMT1 and OOMT2 exhibit different substrate specificities in PME biosynthesis. The main sequence difference is the single amino acid polymorphism in the phenolic substrate binding site. *OOMT1* is mainly found in Chinese roses instead of European roses. It is speculated that *OOMT1* may have evolved from an *OOMT2*-like gene, and its emergence is a critical step in the evolution of scent production in Chinese roses [36].

OOMTs also catalyze the production of methyleugenol and methyl(iso)eugenol by efficiently methylating eugenol and (iso)eugenol in *R. chinensis* [34].

In general, the rose floral scent is dominated by terpenoids, phenylpropanoids, and benzenoids. Only a few fatty acid derivatives are involved in rose floral scents. However, no related enzymes or genes have been isolated or characterized from roses to date.

### 3.3. Transcriptional Regulation of Rose Floral Scent Synthesis

Transcription factors (TFs) have been shown to participate in the coordinated regulation of the scent biosynthetic network [66,67]. Some TFs have been identified for the regulation of floral scents of phenylpropanoids/benzenoids in petunia, including activators of ODO1 [68], EOBI [69,70], EOBII [70,71,72], PH4 [73], and the repressor of PhMYB4 [74]. Recently, only three TFs have been isolated for transcriptional regulation of floral terpenoid production, including GaWRKY1 in *Gossypium arboretum* [75], MYC2 in *Arabidopsis thaliana* [76], and PbbHLH4 in *Phalaenopsis bellina* [77]. Interestingly, some TFs may act upstream of multiple metabolic pathways across terpenoids and phenylpropanoids/benzenoids. When the Arabidopsis transcription factor *production of anthocyanin pigment 1* (*PAP1*) is introduced into the petunia and rose, phenylpropanoid- and terpenoid-derived scent compounds show elevated expressions compared to the control flowers [78,79].

In roses, only one R-type MYB TF, RhMYB1, may play a role in floral scent production, but its function has not been validated [59]. Sun found the expression of *NUDX1* is transcriptionally regulated between scented *R. chinensis* ‘Old blush’ and unscented *R**.×wichurana*, but failed to identify the functionary TFs in the regulation of *NUDX1* [80]. In rose petals, the miR156-*SPL9* regulatory hub is proposed to orchestrate the production of both colored anthocyanins and certain terpenes, by permitting the complexation of preexisting MYB-bHLH-WD40 proteins [15]. The maximum expression of *GDS*, which encodes the enzyme catalyzing germacrene D synthesis, is correlated with miR156 activation and with *SPL9* downregulation. The miR156-*SPL9* regulatory hub might also regulate the expression of terpene synthase genes directly. The absence of the expression of nerolidol synthase (NES) genes is correlated with the downregulation of SPL9 through activation of miR156.

## 4. Regulation of Floral Scent Production and Emission in Roses

### 4.1. Floral Scent Production Sites in Roses

The petal represents the major source of scent in *R. hybrida*. Some scent-related genes are specifically expressed in petals, such as *RcEGS 1*, *RhPAAS*, *rose-PAR,* and *RcPOMT* [52,53,56,57]. Although the cell morphology between the two epidermal layers of the petal is different, both layers are capable of producing and emitting scent volatiles [81,82], which is different from some other species, such as *Antirrhinum majus* [83].

However, not all the scent compounds are produced by petals in roses. In *R. rugosa*, three esters, including citronellyl acetate, geraniol acetate, and nerol acetate, are preferably synthesized in stamens instead of petals [50].

### 4.2. Flower Developmental Stages for Scent Production in Roses

Petal development can be divided into two phases after its initiation in most plants. The first phase is a slow growth stage mainly resulting from cell division, and the second phase is a rapid growth stage resulting only from cell expansion [84,85]. In roses, floral scent production peaks during the second stage, which usually corresponds to a semi-opening state in the flowering process [31]. Changes in major individual compounds are found to be consistent with the changes in total scents during flower development [86]. Several scent-related genes are expressed during this process, such as *RrAAT* [50], *RhPAAS* [52], *rose-**PAR* [53,87], and *OOMT2* [88].

### 4.3. Diurnal Regulation of Scent Production in Rose Flowers

Circadian clock exerts the rhythmic emission of floral scents in numerous species, such as *Nicotiana* [89], *Mirabilis jalapa* [90], *Petunia axillaris* [91], *Vaccinium corymbosum* [92], etc., presumably synchronized with the activity of pollinators. Most floral volatile emissions are regulated by the circadian clock in roses, including the terpenoids geraniol, E-citral, β-cubebene, geranyl acetate, germacrene D [47,93], phenylpropanoids/benzenoids 2-PE, PAA, DMT [52,93,94], and fatty acid derivative hexylacetate [93]. Glycosylated volatiles stored within petals are believed to be the sources of rhythmically emitted volatiles [94].

However, the regulatory effects on scent compounds vary greatly in roses, showing little to no effect on some compounds. In *R. hybrida* flowers, the emissions of several scent compounds, such as oxidized monoterpenols, *trans*-caryophyllene, dihydro-β-ionone, and germacrene D, are completely dependent on light rather than diurnal regulation [47,93], indicating that various independent mechanisms have evolved during diurnal regulation of scent emission in roses.

### 4.4. Environmental and Internal Factors Affecting Rose Scent Production

Floral scent production is sensitive to environmental factors [95]. Among the roses, the cultivar ‘Noorjahan’ of *R. damascene* contains a very different floral aroma complex compared with two cultivars at different altitudes and climatic conditions. A four-fold difference in the main contents of essential oils in cultivars between these two sites has been detected [29].

Light has a great impact on rose floral scent production. In *R. hybrida*, the production of germacrene D and geranyl acetate is directly regulated by light [47]. Ambient temperature is also found to influence scent production in plants. When the temperature increases from 20 °C to 35 °C, the total endogenous amounts of scent components decrease in *P. axillaris* [96]. Under high temperatures in the summer, an alternative principal pathway is induced, through which 2-PE is produced via PPA instead of PAA [55].

Hormones are reported to regulate volatile productions in flowers. GA is found to negatively regulate scent production in petunia flowers by transcriptional/post-transcriptional downregulation of regulatory and biosynthetic scent-related genes [97]. Ethylene is also reported to play a negative role in the floral volatile production in *Petunia hybrida* [98,99] and *Lathyrus odoratus* [100]. However, in one study using 13 *R. hybrida* cut cultivars as materials, scent emission does not appear to be regulated by endogenous or exogenous ethylene but occurs independently of petal senescence or abscission [19].

### 4.5. Internal Mechanisms of Rose Scent Emission after Production

Floral scent composite is controlled by both the endogenous production and emission rate of scent compounds [101]. In roses, not all scent compounds are emitted simultaneously with their production. The emission of germacrene D in *R. hybrida* oscillates during the daily cycle while its endogenous level is constant throughout the day [47]. A similar pattern has been found for 2-PE in *R. damascene* [93].

The tissue structures of flowers do not correlate with floral scent emissions in roses, as shown by petal anatomy, which indicates there is no significant difference in optical or ultrathin sections between scented and unscented rose cultivars [82].

The endogenous concentrations of steady-state volatiles are correlated with floral scent emissions in plants [102]. Floral scent compounds, such as 2-PE, geraniol, and benzyl alcohol are present in flower tissues in the forms of monoglycosides and/or diglycosides. After being hydrolyzed by β-glucosidase or endoglycosidase, these glycoconjugates become volatile and emit from petal tissues [103,104,105,106]. Moreover, 2-PEG is a source for the circadian-emitted 2-PE in *R. damascene* [93]. β-glucosidase is involved in the emission of 2-PE from rose flowers by hydrolyzing 2-PEG [107].

Passive diffusion has been assumed to be the mechanism for floral scent emission. However, studies have shown that exocytosis or specific transporters may be involved in the transport of scent molecules across the plasma membranes [108]. Recently, an adenosine triphosphate-binding cassette (ABC) transporter rather than passive diffusion has been shown to be responsible for floral volatile emission in *P. hybrida*. Down-regulation of the ABC transporter gene *PhABCG* results in decreased emission of volatiles and increased toxic accumulation in the plasma membrane [109]. As the petal develops, OOMTs are increasingly associated with membranes, suggesting the involvement of the secretory machinery of cells in scent emission in roses [33].

## 5. Molecular Breeding and Metabolic Engineering of Rose Floral Scents

Floral volatiles could respond rapidly to artificial selection [110]. However, most modern cultivars used for cut flowers have little fragrance as a result of breeding preference for other traits, such as the flower color and form or longevity in roses [35]. Little is known about the inheritance of rose scents.

Three volatiles, neryl acetate, geranyl acetate, and nerol, show monogenic or oligogenic traits in the process of sexual hybridization of *R. hybrida*, while three other volatiles, 2-PE, geraniol, and β-citronellol, show quantitative inheritance. All three monogenic volatiles and six QTLs of the three quantitative volatiles have been mapped on the rose chromosome map, together with some scent-related genes for germacrene D, alcohol acetyltransferase 1, and various OMTs [46,111]. To date, dozens of scent-related genes or traits have been mapped to the linkage group (LG) (Table 3) [46,111,112,113,114,115,116].

However, it is still far from rationally constructing scent attributes by crossbreeding or molecular approaches. A hybrid tea progeny has been obtained from parents emitting very different floral scents. As a general rule, the quantity and quality of volatile compounds are both low in progeny members [117]. Many uncertainties may be involved in rose scent inheritance, such as gene regulation or mutation, substrate availability, and substrate preference of enzymes [51,78].

It is possible to produce rose scents in model plants, such as petunias. However, in order to regulate the metabolic flow of rose scent compounds, a detailed analysis of available substrates and enzyme activities in petunias is required as a prerequisite [51]. Additionally, yeast has been shown to be another carrier for metabolic engineering of rose scent biosynthesis [52].

## 6. Obstacles in Further Study for Rose Floral Scents

Floral scents in modern roses are complex combinations of scents from ancient Chinese roses and European roses. Understanding the floral scents can be used to generate the genealogy for *Rosa* cultivars at the genus level [27,118]. The past decade has witnessed the development of various approaches and tools for molecular studies of rose scents. Metabolic profiling and gene expression profiling are powerful tools for the identification of candidate genes and enzymes in scent synthesis [18,119]. However, some obstacles still hinder the in-depth understanding of the molecular mechanisms of floral scents in roses.

Functional validation of candidate genes is largely limited in roses because no universal transformation system is available. To date, the functions of most rose genes are validated in heterologous systems, such as the petunia, *Escherichia coli*, and yeast [31,32,33,51,52]. Virus-induced gene silencing (VIGS) is a promising approach for gene knockdown in roses, despite its efficiency remaining low compared to tobacco and tomato [44,120]. Furthermore, an effective method of “graft-accelerated VIGS” was developed especially for research on flower characteristics, such as floral scent, floral color, and so on [121]. Currently, it is still not enough to predict the substrate preference of candidate enzymes in roses based on gene homology with other species, because protein sequence divergence may affect enzyme affinity to the substrates [51,122]. Moreover, the preferred substrates of a candidate enzyme in in vitro assays or heterologous systems may not necessarily be the ones in vivo [32,51,122].

Studies based on the next-generation sequencing (NGS) technology have been reported, including various transcriptome studies, providing a wealth of information about the expression patterns of genes or miRNAs in multiple tissues or biological processes in roses [42,123,124,125,126,127]. A draft genome sequence of a wild diploid rose (*R. multiflora* Thunb.) has been generated, which provides an important resource for fundamental studies and applications of roses [128]. By generating a homozygous genotype from a heterozygous diploid modern rose progenitor, *R. chinensis* ‘Old Blush’, the most comprehensive rose genome, is created through single-molecule real-time sequencing and a meta assembly approach [15]. This genome provides a foundation for understanding rose traits. By reconstructing regulatory and secondary metabolism pathways, the regulation of scent is proposed to be interconnected with that of the flower color. However, it still has limitations in studies of cultivated roses because most of the roses are tetraploids.

In crossbreeding for rose scents, one of the major hindrances to creating novel varieties stems from the lack of allelic variations in modern roses, because their genome bears a massive introgression of *R. chinensis* alleles [129]. Another hindrance is the polyploidy barrier, which appears when alleles of interest are transferred from wild diploid species to tetraploid modern roses [20]. It has been found that manipulating temperature may be a strategy to overcome the polyploidy barrier [130]. The third hindrance arises from the limitations of genetic mapping. In these studies, due to the inheritance complexity of tetraploid populations, most genetic mappings for locating or tagging scent-related genes are conducted using diploid populations [131,132,133,134], which causes great difficulties when localizing scent characteristics in crossbreeding, because most modern roses are tetraploid.

Our inadequate understanding of odor metabolism pathways greatly limits the genetic engineering of the floral scent-related traits of roses. When a scent enzyme is newly introduced, we need to consider whether appropriate substrates can be found and whether enough products can be produced. Transcriptional factors seem to provide an efficient strategy to manipulate metabolic flux flow for rose scent modification [68,78,79]. Overall, it is clear that genetic manipulation of the rose floral scent is possible but requires a more rational design.

## 7. Conclusions and Future Perspectives

Over the last two decades, an increasing amount of knowledge on the biosynthesis of rose floral scents has been acquired. Recent advances in the isolation and characterization of genes and enzymes involved in different scent biosynthetic pathways have enhanced our understanding of floral scent synthesis in the rose. Despite the progress, many aspects remain largely unknown. In particular, we still do not know the regulatory networks controlling the synthesis pathways of rose floral scents, including how the majority of floral scents are synthesized and how their orchestrated emissions are regulated, which hinder their molecular breeding and metabolic engineering. Therefore, we anticipate that future research efforts will focus on exploring the regulatory mechanisms of rose floral scents; meanwhile, some more advanced approaches should be developed and applied for elucidating the tetraploid rose genome and function validation of candidate genes.

## Figures and Tables

**Figure 1 ijms-23-08014-f001:**
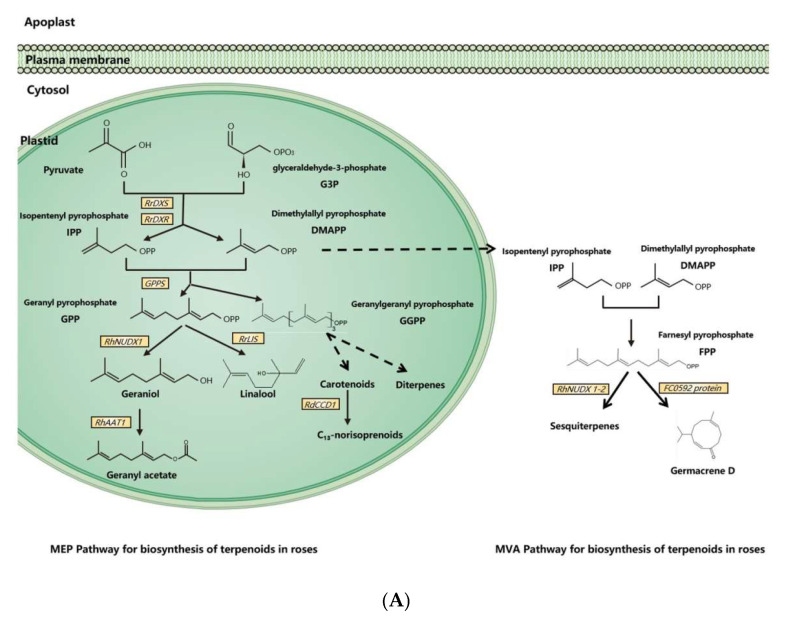
Synthesis pathways of floral scents in roses. (**A**) The biosynthesis pathway of terpenoids in roses. (**B**) Biosynthesis pathway of phenylpropanoids/benzenoids in roses. Solid lines indicate established biochemical reactions, and broken lines indicate possible steps. G3P: glyceraldehyde-3-phosphate; DXS: 1-deoxy-D-xylulose-5-phosphate synthase; DXR: 1-deoxy-D-xylulose-5-phosphate reductoisomerase; IPP: isopentenyl pyrophosphate; DMAPP: dimethylallyl pyrophosphate; GPP: geranyl pyrophosphate; GPPS: GPP synthase; GGPP, geranylgeranyl pyrophosphate; NUDX: nudix hydrolase; LIS: linalool synthase; AAT: alcohol acetyltransferase; CCD: carotenoid cleavage (di-)oxygenase; FPP: farnesyl pyrophosphate; PAL: phenylalanine ammonia lyase; PAAS: phenylacetaldehyde synthase; AADC: aromatic amino acid decarboxylase; AAAT: aromatic amino acid aminotransferase; PPA: phenylpyruvic acid; PAA: phenyl acetaldehyde; PAR: phenyl acetaldehyde reductase; PPDC: phenylpyruvate decarboxylase; 2-PE: 2-phenylethanol; POMT: phloroglucinol O-methyltransferase; OMT: O-methyltransferases; OOMT: orcinol O-methyltransferases; MHT: 3-methoxy-5-hydroxytoluene; DMT: 3,5-dimethoxy toluene; DHA: 3,5-dihydroxy antisole; DMP: 3,5-dimethoxy phenol; TMB: 1,3,5-trimethoxy benzene.

**Table 1 ijms-23-08014-t001:** Major components of rose floral scents.

Compound Variety	Compounds	Odor	References
terpenes	β-cubebene	citrus, fruity, radish	[25,31]
	β-elemene	herbal, waxy, fresh	[26,31]
	δ-cadinene	thyme, herbal, woody	[27]
	germacrene D	woody, spice	[31]
	geraniol	rose-like, sweet	[28,31,32]
	citronellol	fresh rosy	[27,28,31]
	nerol	lemon-like, floral	[27,28,32]
	linalool	citrus and floral	[27,32]
	farnesyl acetate	green-floral rose	[27]
	geranyl acetate	lavender	[27,31]
	citronellyl acetate	fresh, rose, fruity odor	[25,31]
	neryl acetate	rose and lavender-like	[25,31]
	citral	citrus and lemon	[25,27]
	dihydro-β-ionone	violet-like and earthy	[27]
	rose oxide	herbal, green floral, earthy	[30]
Phenylpropanoids/ benzenoids	2-phenylethanol	honey-like	[28,30,31]
	2-phenylethyl acetate	sweet, honey, rosy, with a slight yeasty honey note	[27,30,31,32]
	1,3,5-Trimethoxybenzene (TMB)	phenolic spicy, earthy note	[28,33,34]
	dimethoxytoluene (DMT)	fresh, earthy, phenolic spicy	[28,33]
	benzyl acetate	floral, fruity, sweet, fresh	[27,32]
	eugenol	clove, carnation	[28,35]
	methyl eugenol	clove, carnation	[27,34]
	methyl isoeugenol	clove, carnation, woody	[27,34]
fatty-acid derivatives	*cis*-3-hexenyl-1-alcohol	fresh and leafy green	[25,28,32]
	2-hexenyl acetate	fresh, fruity green	[27,31,32]
	*cis*-3-hexenyl acetate	fresh and leafy green	[27,31]

**Table 2 ijms-23-08014-t002:** Scent-related genes in roses.

Pathway	Gene	Species	References
Terpenoids	*RhGDS*	*R. hybrid* ‘Fragrant Cloud’	[47]
	*RhCCD1*	*R. damascene*	[48]
	*RhCCD4*	*R. damascene*	[49]
	*RrLIS*	*R. rugosa* Thunb. ‘Tangzi’	[50]
	*RhAAT*	*R. hybrida* ‘Fragrant Cloud’	[32,51]
	*RrAAT*	*R. rugosa* Thunb. ‘Tangzi’	[50]
	*RrDXS*	*R. rugosa* Thunb. ‘Tangzi’	[50]
	*RrDXR*	*R. rugosa* Thunb. ‘Tangzi’	[50]
	*RcGDS*	*R. chinensis* ‘Old Blush’	[42]
	*RhNUDX1*	*R. chinensis* ‘Old Blush’	[18]
Phenylpropanoids /benzenoids	*RhPAAS*	*R. hybrida* ‘Fragrant Cloud’	[52]
	*Rose-PAR*	*R.×damascena* Mill	[53]
	*RyAAAT3*	*R. hybrida* ‘Yves Piaget’	[54]
	*RyPPDC*	*R. hybrida* ‘Yves Piaget’	[55]
	*RcPOMT*	*R. chinensis* Jacq.var. *spontanea*	[56]
	*RcEGS1*	*R. chinensis* ‘Old Blush’	[57]
	*RcOOMT1*	*R. hybrida* ‘Fragrant Cloud’ and ‘Golden Gate’; *R. chinensis* Jacq.var. *spontanea*	[34,35]
	*RcOOMT2*	*R. hybrida* ‘Fragrant Cloud’ and ‘Golden Gate’; *R. chinensis* Jacq.var. *spontanea*	[34,35]
	*RcOMT 1*	*R. chinensis* var. *spontanea*	[34]
	*RcOMT 2*	*R. chinensis* var. *spontanea*	[34]
	*AADC*	*R.* ‘Hoh-Jun’	[58]
	*RhMYB1*	*R. hybrida* ‘Jinyindao’	[59]

**Table 3 ijms-23-08014-t003:** Phenotypic and genotypic traits, available on the consensus map in roses.

Pathways	Scent-Related Genes or Traits	LG	Population	Rose Species	References
terpenoids	geraniol (QTL)	LG1	Linkage groups 94/1	*R. multiflora*	[114]
	TPS-L (Terpene synthase-like)	LG1	Linkage groups 94/1	*R. multiflora*	[46]
	*RhCCD1*	LG1	Linkage groups 94/1	*R. multiflora*	[46,114]
	*RhAAT1*	LG2	Linkage groups 94/1 + Linkage groups 97/7	*R. multiflora*	[46,114]
	geranyl acetate	LG2	Linkage groups 94/1	*R. multiflora*	[114]
	nerol	LG3	Linkage groups 94/1	*R. multiflora*	[114]
	β-citronellol (QTL)	LG3	Linkage groups 94/1	*R. multiflora*	[114]
	neryl acetate	LG4	Linkage groups 94/1	*R. multiflora*	[114]
	TPS-L (Terpene synthase-like, Farnesyltransferase)	LG4	Linkage groups 94/1	*R. multiflora*	[46]
	GDS	LG5	Linkage groups 94/1 + Linkage groups 97/7	*R. multiflora*	[46,114]
	TPS-L (Terpene synthase-like)	LG5	Linkage groups 94/1	*R. multiflora*	[46]
	alcohol acetate	LG7	Linkage groups 94/1	*R. multiflora*	[114]
Phenylpropanoids	RhPAR	LG1	Linkage groups 94/1	*R. multiflora*	[46,114]
	*RhOOMT1*	LG2	Linkage groups 94/1	*R. multiflora*	[114]
	*RhOOMT2*	LG2	Linkage groups 94/1	*R. multiflora*	[46]
	RcOMT3-1	LG2	Linkage groups 94/1 + Linkage groups 97/7	*R. multiflora*	[114]
	RcOMT3-265	LG2	Linkage groups 94/1	*R. multiflora*	[46]
	BEAT-L	LG2	Linkage groups 94/1	*R. multiflora*	[46]
	NMT-L (N-methyltransferase)	LG2	Linkage groups 97/7	*R. multiflora*	[46]
	BEAT-L	LG4	Linkage groups 97/7	*R. multiflora*	[46]
	*RcOMT3*-*2*	LG4	Linkage groups 94/1 + Linkage groups 97/7	*R. multiflora*	[114]
	*RcOMT3*-280	LG4	Linkage groups 97/7	*R. multiflora*	[46]
	*RcOMT1*	LG4	Linkage groups 97/7	*R. multiflora*	[46,114]
	phenylethanol (QTL)	LG5	Linkage groups 94/1	*R. multiflora*	[114]
	*RhAADC*	LG5	Linkage groups 97/7	*R. multiflora*	[114]
	POMT	LG6	Linkage groups 94/1 + Linkage groups 97/7	*R. multiflora*	[46,114]

## Data Availability

Not applicable.

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
