# Peer review of "Genetic and Biochemical Aspects of Floral Scents in Roses"

_ijms, 2022, doi:10.3390/ijms23148014_

Round 1

Reviewer 1 Report

1. The authors need to enrich the abstract by adding a concentrated concept of the main theme and recent key achievements in this field.

2. It's strange to put a section of discussion with a conclusion in a review article. Commonly, a conclusion can combine perspectives and it needs to avoid citing references in this part because the authors have reviewed literature and have to give a digested and concentrate content for the conclusion, and also give some future directions for this research field.

3. L19: Floral scent - The floral scent

4. L33: the second largest - the second-largest

5. L96: the original oil bearing cultivars - the original oil-bearing cultivars

6. L339: was developed specially for researches to - was developed especially for research on

7. to create novel varieties - to creating novel varieties

8. After reading this manuscript, I found it is very valuable for this research field and can be accepted for publication in IJMS.

Reviewer 2 Report

Line 20: add a space after “pollinators”, check similar throughout the manuscript.

Line 43: industry [15]

Lines 51-53: Check the grammar of the sentence: “Unraveling the molecular bases of flower scents is not only a fascinating topic in the rose biology, but will improve rose production in the floriculture industry.”

Line 68: delete the word “cultivars”, these are species

Line 69: delete the word “cultivars”, these are species

Line 78: DMT [36,37]

Line 312: Table 3: suggest changing “Rose Variety” to “Rose Species”

Round 2

Reviewer 1 Report

It has been revised accordingly.